# Evaluating Remote Task Assignment of an Online Engineering Module through Data Mining in a Virtual Communication Platform Environment

Zoe Kanetaki [1,*], Constantinos Stergiou [1], Georgios Bekas [2], Christos Troussas [3] and Cleo Sgouropoulou [3]

1 Department of Mechanical Engineering, University of West Attica, 12241 Egaleo, Greece; stergiou@uniwa.gr
2 Department of Civil Engineering, University of West Attica, 12241 Egaleo, Greece; gb331984@googlemail.com
3 Department of Informatics and Computer Engineering, University of West Attica, 12241 Egaleo, Greece; ctrouss@uniwa.gr (C.T.); csgouro@uniwa.gr (C.S.)
* Correspondence: zoekanet@uniwa.gr

**Abstract:** E-learning has traditionally emphasised educational resources, web access, student participation, and social interaction. Novel virtual spaces, e-lectures, and digital laboratories have been developed with synchronous or asynchronous practices throughout the migration from face-to-face teaching modes to remote teaching during the pandemic restrictions. This research paper presents a case study concerning the evaluation of the online task assignment of students, using MS Teams as an electronic platform. MS Teams was evaluated to determine whether this communication platform for online lecture delivery and tasks' assessments could be used to avoid potential problems caused during the teaching process. Students' data were collected, and after filtering out significant information from the online questionnaires, a statistical analysis, containing a correlation and a reliability analysis, was conducted. The substantial impact of 37 variables was revealed. Cronbach's alpha coefficient calculation revealed that 89% of the survey questions represented internally consistent and reliable variables, and for the sampling adequacy measure, Bartlett's test was calculated at 0.816. On the basis of students' diligence, interaction abilities, and knowledge embedding, two groups of learners were differentiated. The findings of this study shed light on the special features of fully online teaching specifically in terms of improving assessment through digital tools and merit further investigation in virtual and blended teaching spaces, with the goal of extracting outputs that will benefit the educational community.

**Keywords:** CAD; data analysis; data mining; online learning; engineering education; COVID-19; MS Teams

## 1. Introduction

Web-based education is centred on the use of Learning Management Systems (LMSs) and communication platforms to ensure that all members of the academic society have access to learning resources in the absence of physical teaching spaces. Synchronous or asynchronous communication is provided, via asynchronous system tools such as e-mail, chat features, and learning newsgroups [1]. It is crucial to examine major aspects in online learning that reflect each method's efficacy. Those aspects include indicators of how e-learning platforms respond to the demands of students and their instructors, provide distant instruction on effectively performing learning activities, and whether exclusive online classes can become a sustainable tool for time periods when online learning takes place entirely remotely [2].

Engineers' professional activities are different from others, since they design objects that do not exist in nature aiming to ameliorate the way of living [3]. Engineering requirements, on the other hand, have resulted in the development of new teaching methods at all levels of the educational community. Novel technological systems, such as LMSs, as

well as online media channels, are integrated into the educational process and serve as the principal method of transferring new knowledge [4–9]. Applying learning strategies carefully structured and based on Moore's principles [10] of transactional distance, previous research indicated that numerous module features can be adjusted in order to improve the framework, communication, and self-confidence during the educational process. Moreover, researchers investigated ways to make recommendations based on users' location, rating history, and suggestion coverage. A multi-objective recommendation system was developed to tackle the challenge of user preferences and suggestion coverage [11]. Based on a multi-population technique presented in [12], a differential evolutionary algorithm is implemented in each population and harvests a multi-population multitask differential evolutionary optimisation. In [13], an Emergency Remote Teaching Environments (ERTE) was introduced, while in [14], new elements affecting online learning were discovered. The involvement of social media channels for asynchronous support, the evaluation of the presentation modes of the e-module content, the impact of technical issues occurring during and beside the synchronous lectures, and the time needed to fulfil the assignments combined with the students' individual perception of the weekly task load were analysed in [13]. In [15], the application of mobile social networks in the educational process was researched, and significant correlations were brought to light concerning the influence of students' satisfaction towards MNSs. Researchers have also investigated the role of mobile learning applications in online education during the pandemic and have linked the role of technical features, as well as individual factors with student's emotional state [16]. A thematic analysis technique using NVivo software was performed by investigating factors that affect the adoption of e-learning systems, as well as the challenges faced when adopting an e-learning system in Jordanian Universities [17]. Researchers proposed and tested a teaching method for an engineering CAD module in higher education during the pandemic, aiming to develop learners' spatial conception and insert real-world tasks of mechanical engineering into a virtual classroom [18]. The use of a single platform for real-time lecture transmission, task assessments, gradings, and asynchronous support was tested through statistical analysis, and the results showed that a communication platform used for all the above purposes can assist the educational procedure [19]. A comparison between the parallel use of the MS Teams communication platform with Moodle as a primary LMS and the use of MS Teams for all educational purposes including task assignments was investigated in [20], and even though no direct impact was found on students' academic performance, the use of multiple LMSs resulted in higher levels of dropout cases. Learners' ability to complete assignments, the contribution of the theoretical part of the e-lecture, the quizzes and the assignments' variety, and classroom fatigue were some of the 24 variables shown in [18] to be predictors of students' satisfaction. In [21], the researchers focused on conceiving the impact of technology on teaching and learning methods, as well as the ability to adapt and respond to changes and opportunities by digital means. In [22], the researchers investigated the criteria of the assignments' content, in order to distinguish which could be virtualised following a return to normality, from those that could be performed in face-to-face learning spaces.

Nonetheless, the first academic semester of 2021–2022 is undoubtedly a watershed period when referring to the educational procedure. The time period of the disruption caused by the COVID-19 pandemic has not yet come to an end, and educational models that have been applied during this time span will be affected during this moment of the return to "normality" [23].

Institutions and instructors worldwide are facing a dilemma related to which of the existing components of the ERTE [13] will be maintained and which will be abandoned [23]. For this purpose, remote task assessment methods that have been applied during the pandemic need to be analysed, taking into consideration the specific context of the restrictions that took place during the specific period [22].

This research was motivated by the growing usage of online-based education in institutions around the world, which offers a unique chance to collect electronic data and monitor students' educational performance.

The novelty of this study stems from the fact that pandemic constraints in educational institutions were just recently placed, and fully virtual learning environments are yet to be explored.

The current study evaluates the learning strategy of an online first-year engineering course from the perspective of students, during a public health emergency period in the context of evaluating task assignment tools via a single learning platform. By utilizing data from online surveys, a statistical analysis was conducted that identified new aspects with significant correlation to the way students experience task assignments during a pandemic.

## 2. Materials and Methods

The current research was carried out in the Department of Mechanical Engineering, School of Engineering, University of West Attica, during the first semester of the academic year 2020–2021. The associated module is titled Mechanical Design—CAD I.

### 2.1. Module Content and Learning Goals

The learning goals of this first-semester module are for students to be able to produce 2-dimensional sketches and Computer-Aided Design (CAD) drawings of elements shown in 3-Dimensional views, while following the rules of mechanical drawings. Conceiving the form of a 3-Dimensional object, being able of exploding it and representing it in different views (top view, side view, sectional view) are the primary teaching goals of the module. The basic mechanical design rules consist of the placement of views in the paper layout, symbolizing hidden edges with dashed lines and symmetry with dashed-dotted lines or long dashed lines, as well as defining hatched areas within sectional views.

The laboratory lectures lasted four months and were conducted exclusively online using the Microsoft Teams platform, due to the pandemic restrictions. Due to the module's context (free-hand drawings and CAD drawings), the first 4 online lectures could be easily attended using any mobile device, including smartphones and tablets. Sketches drawn by students and asynchronous support and quizzes performed in online MS Forms could be uploaded directly to the communication platform by smartphone, without the use of a PC or laptop. However, from the 5th online lecture on, a laptop or personal computer was needed in order to perform tasks through a computer-aided design program, offered in an educational version for PC applications.

### 2.2. Description of the Assignments

Quizzes, practice problems, i.e., non-existent 3-dimensional objects that were to be represented in 2-Dimensional drawings, and learning tasks related to real-world activities in Mechanical Engineering [24–26] were uploaded in the form of assignments during each synchronous lecture (Figure 1). The nature of the assignments was in accordance with the presentation of new knowledge. A total of 22 tasks were assigned and graded during the semester, starting from recalling previous knowledge and proceeding to the stage of assimilating new knowledge by practice.

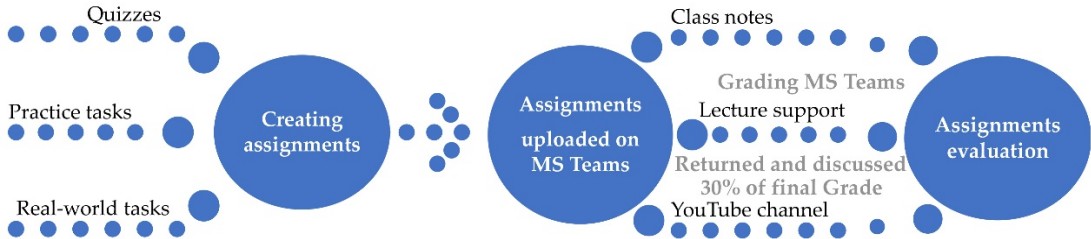

**Figure 1.** Scheme of the tasks' assignment procedure.

The objects aimed to be studied were initially presented in a 3-dimensional drawing, in a single isometric view. As the objects started to become more complicated in terms of form, 3-dimensional models were constructed in the Autodesk Inventor CAD program, while videos of those objects in rotation were created, aiming for all object views to be visible and clear, as seen in Figure 2. The videos were uploaded to the MCAD I UNIWA YouTube channel, and their links were attached as reference material in the related task assignment, as shown in Figure 3. Specific instructions about the detailed steps required to accomplish the tasks and the software required were also uploaded in each task assignment. A thumbnail can be seen in Figure 4.

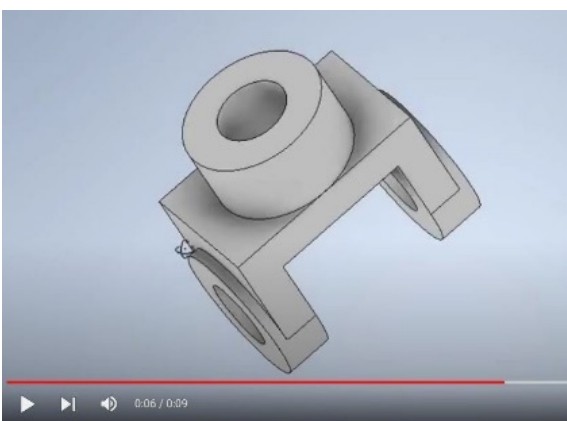

**Figure 2.** Rotated object.

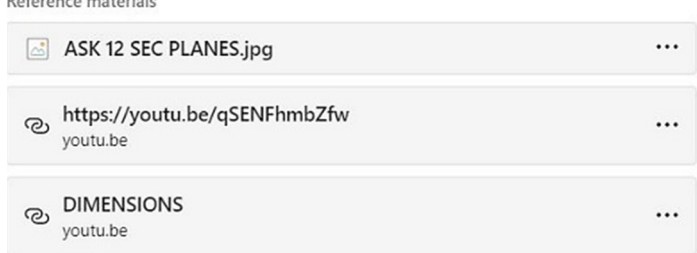

**Figure 3.** Reference materials.

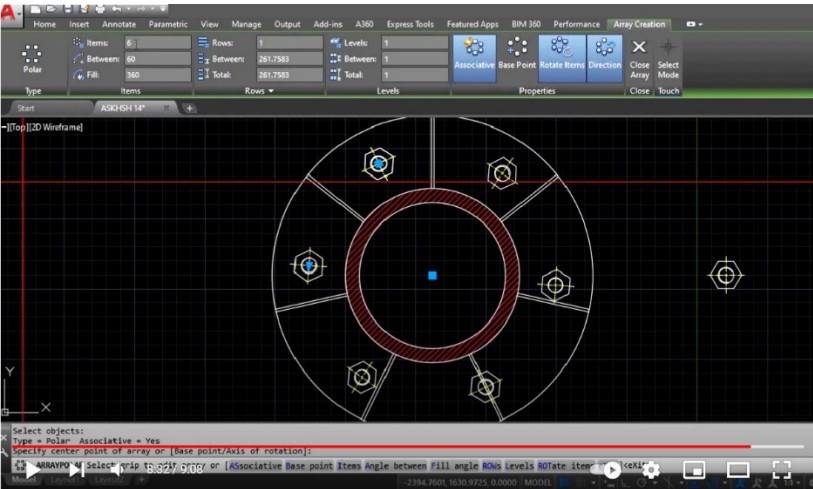

**Figure 4.** Software instructions.

### 2.3. Data Mining Sources

The information used for this research was gathered from several sources. Among them, there were students' registry data, which provided information about students' previous educational performance, such as their grades on the university level entrance exams in four subjects: physics, mathematics, chemistry, and dissertation. Each student's high school type was considered, between general lyceum and vocational lyceum. Students' activity reports were provided by the MS Teams communication platform, which included a median of their assignments' grade, virtual presence, and asynchronous presence on the platform [19,20,27]. An organogram of the data collection can be seen in Figure 5.

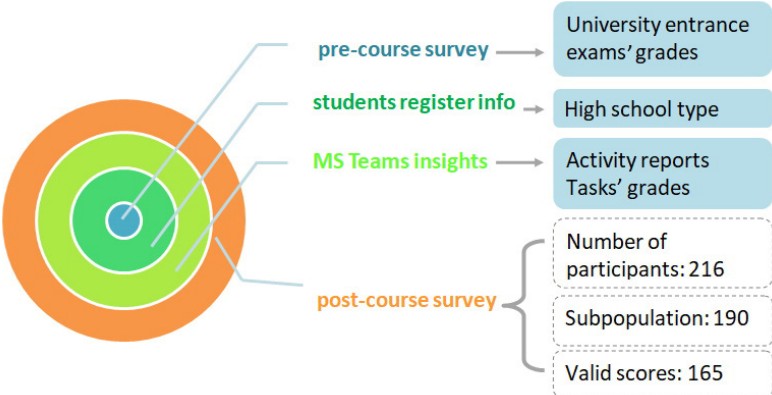

**Figure 5.** Organogram of the data collection method.

The pre-course questionnaire was addressed to the learners during their registration in the university module, in order to obtain information about their previous performance, on the university entrance exams. The post-course survey was launched during the last online lecture. The purpose of this questionnaire was to investigate the impact of several constructs related to the online learning process during the pandemic, e.g., demographic factors, technical difficulties, the impact of social distancing, e-course presentation, and computer skills, on students' learning achievements [13]. Among other constructs, the construct of "assignments" was identified through 37 survey questions [19]. All answers were associated with the registry number of each student, in order to link their attitude towards the online module with their past academic performance. There were 216 students initially registered in the online module, divided into 11 online teaching groups. There were 190 students out of 216 who participated in the survey. Out of those 190 participants, 165 scores were retained and considered valid for this research, since the remaining 25 scores reflected older attendants, not first-year students. The configuration of the related sections of the questionnaire can be seen in Figure 5.

### 2.4. Data Sample Characteristics

In order to understand the characteristics of the sample under the specific context of the pandemic, demographic factors and additional social aspects were included in the questionnaire, as described below:

Out of the 165 participants, most of them (142 students) were aged between 18 and 21 years old (86.06%). Of these, 10.30% (17 students) were between 22 and 25 years old, and 2.42% (4 students) were between 26 and 29 years old, while 0.60% (2 students) were between 34 and 37 years old. The majority were male students (88%) and the remaining 12% female students [16]. There were 64% attending the e-lectures from the same town, 34% from other cities or provinces, and 2% from other countries. There were 77% living in the same house with their families, 14% living alone, 4% living with their siblings, 4% living with their companions, and the remaining 1% living with a friend. In June 2020, students passed the University entrance exams, with an average grade in mathematics of 13.90, in essay pf 13.20, in chemistry of 12.40, and in physics of 13.95. As regards the selection of

the University Department of Mechanical Engineering, 64% stated that it was their first choice, versus 33% who selected the department due to the proximity of the campus to their place of residence. The remaining 3% stated that their department selection was based on other criteria.

*2.5. Data Filtering Methods*

The constructs previously mentioned, enriched with additional learners' information, as seen in Figure 5, led to the creation of a matrix of 129 variables corresponding to the survey questions, registry information, and platform activity reports, multiplied by 165 answers from first-semester students (considered valid after eliminating the participants of higher semesters). An MS Excel file was created after exporting the MS Form questionnaire, which was enriched by the other two sources [19].

An SPSS sav file was formed containing 129 variables, by inserting the Excel data sheet. There were 37 variables isolated into a single SPSS sav file, by conducting a correlation analysis, which brought to the surface significant correlations among the ordinal variables. A threshold of >0.20 was set in the calculation of Spearman's rho coefficient, enabling the detection of strong relationships between two variables. An MS Excel file was exported, and by isolating relevant data, strong correlations were discovered, as can be seen in Tables 1–5. An organogram of those correlations is shown in Figure 6.

**Table 1.** Variables highly correlated with the module's enjoyability.

| | |
|---|---|
| Platform familiarisation | 0.432 |
| Quality of videos | 0.421 |
| Lower classroom fatigue | 0.475 |
| Succeeded in similar future tasks | 0.518 |
| Overall evaluation | 0.518 |

**Table 2.** Variables highly correlated with the module's organisation.

| | |
|---|---|
| Overall evaluation | 0.532 |
| Evaluation of class notes | 0.448 |
| Tasks' assignment | 0.416 |
| Enjoyability | 0.444 |

**Table 3.** Variables highly correlated with the 15th assignment.

| | |
|---|---|
| Relevant to future professional tasks | 0.513 |

**Table 4.** Variables highly correlated with the quality of videos.

| | |
|---|---|
| Presenting tasks' methodology | 0.426 |
| Enjoyability | 0.421 |
| Variety of tasks | 0.426 |

**Table 5.** Variables highly correlated with understanding 3-dimensional planes.

| | |
|---|---|
| Succeeding in similar future tasks | 0.474 |
| Evaluation | 0.466 |

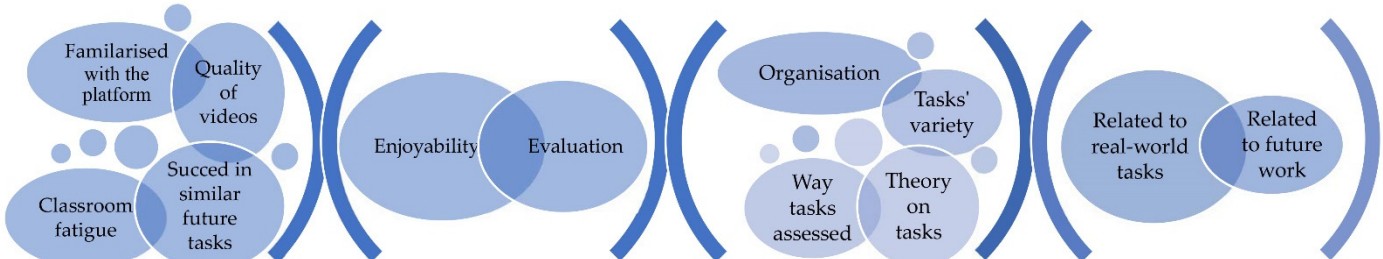

**Figure 6.** Organogram of the correlation among nominal variables.

*2.6. Difficulties Encountered during the Research*

Several difficulties were encountered during this research. Both questionnaires were identified, and some students that filled in the pre-course survey quit the module before having the opportunity to participate in the post-course questionnaire. In addition to that, the second survey contained multiple variables aiming to investigate several constructs. The total number of questions of the second survey was 109, and the average time to complete was 34:56, which implied a significant level of difficulty. Both surveys needed to be fused into one, by incorporating data from the student registry, one by one. Nevertheless, students were overall highly motivated to participate in both surveys, probably due to the lockdown measures and the students' need to express their opinion and attitudes towards the specific learning method.

### 3. Results and Discussion

*3.1. Correlation Analysis Outcomes and Discussion*

In Table 1, the construct of enjoyability is highly correlated with more specific parameters, such as platform familiarisation (MS Teams), quality of videos for asynchronous support, lower classroom fatigue, and higher evaluation of the module. In Table 2, the organisation of the online module is highly correlated with its overall evaluation, the evaluation of the class notes, the tasks' assignment, and enjoyability. In Table 4, the 15th assignment is isolated [18–20,27], since it refers to the study of an existing building structure, located on the University Campus, which students had not yet visited due to the pandemic restrictions and curfew. Specific elements of this structure include metallic details such as bolts, flanges, neurons, and welding, which are essential components of mechanical engineering parts. It has been proven that this assignment is relevant to future work and professional tasks. In Table 4, the quality of videos is related to the task's methodology, the task's variety, and the enjoyability. Finally, regarding Table 5, the level of understanding of 3-dimensional planes was researched and found to be highly correlated with the overall evaluation of the module and students' attitudes towards succeeding in similar future tasks.

The qualitative analysis of the correlations can be seen in Figure 6.

*3.2. Reliability Analysis, Outcomes, and Discussion*

Cronbach's alpha coefficient was determined as an assessment of the internal consistency of the questionnaire, determining the degree of interlinking among the values derived from the analysis, while a reliability analysis was undertaken. Cronbach's alpha coefficient was the result of the reliability analysis undertaken. The variables used for the calculation of Cronbach's alpha coefficient were the ordinal variables, which refer to the survey questions where the answers had to be formed in an ordinal way, using the same scale in terms of the measured objective. With this calculation, the validity of those 37 variables was examined, since they were isolated out of 129 from the correlation analysis, as shown in detail in Section 3.1.

As shown in Table 6, Cronbach's alpha coefficient exhibited that the nominal variables (from the questionnaire) were considered internally reliable variables to an extent equal to 89.1%. Since Cronbach's alpha usually ranges from 0–1, 0.891 is considered a good measurement of reliability. It can be noticed in Table 6 that one student was excluded

from the test (one-hundred sixty-four valid out of one-hundred sixty-five), since there was missing information from the database. 37 variables were proven statistically to be inter-nally consistent. It can be noticed from the calculation of Cronbach's alpha of the 33rd variable (0.887) that if it were deleted, the overall coefficient would drop by 0.004. Nevertheless, the 33rd variable is considered as a highly significant variable in terms of relating virtual classroom assessments to real-world tasks in mechanical engineering, so this particular survey question was not eliminated from the analysis.

**Table 6.** Case Processing Summary and Reliability Statistics.

|  |  | N | % | Cronbach's Alpha | Cronbach's Alpha Based on Standardized Items | N of Items |
|---|---|---|---|---|---|---|
| Cases | Valid | 164 | 99.4 | 0.891 | 0.894 | 37 |
|  | Excluded [a] | 1 | 0.6 |  |  |  |
|  | Total | 165 | 100.0 |  |  |  |

[a] Listwise deletion based on all variables in the procedure.

### 3.3. Sample Adequacy Measure Outcomes and Discussion

Two statistical tests were performed in order to measure the sample adequacy:

The Kaiser–Meyer–Olkin test and Bartlett's test of sphericity were performed, where the first was given a value equal to 0.816, proving that the sample had good adequacy. The result from Bartlett's test entailed that the highest significance (0.000) was attained, as shown in Table 7.

**Table 7.** Variables KMO and Bartlett's test.

| **Kaiser-Meyer-Olkin Measure of Sampling Adequacy.** |  | **0.816** |
|---|---|---|
| Bartlett's Test of Sphericity | Approx. Chi-Square | 2166.018 |
|  | df | 667 |
|  | Sig. | 0.00 |

### 3.4. Cluster Analysis and ANOVA Analysis Performed on the Clusters

A cluster analysis was computed, by the use of the K-means algorithm in SPSS. The population was—therefore—segregated into two learner groups; each group's special characteristics are shown in Figure 7. The determining variables as derived from clustering, which involved two main groups, such as enjoyability, satisfaction, insecurity, technical issues, perception of workload, organisation, and expression, are summarised in the scheme of Figure 7.

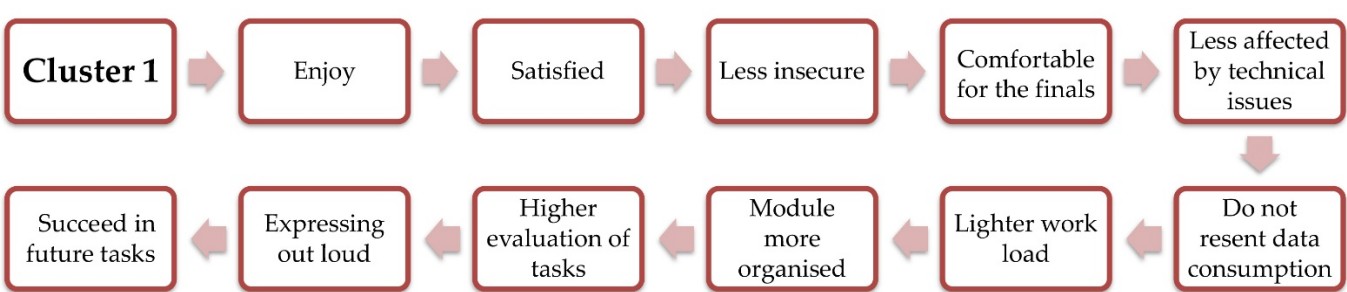

**Figure 7.** Characteristics of the differentiation of Cluster 1.

It can be deduced that categorizing the participants into two clusters was based on the learners' dedication to the e-module, their abilities, and the level of knowledge retention, allowing them to feel that they will be able to successfully perform a comparable future project.

In Tables 8 and 9, the cluster analysis tables demonstrate the most significant criteria of the cluster analysis. In the last column of Table 9, the ANOVA field shows the level

of the variables' significance, which for most of them was <0.05. The two variables with an ANOVA significance >0.05 were students' satisfaction with the grades of their weekly assignments (0.28) and tasks related to their future employment (0.23). It can be deduced that those two variables can be regarded as less representative of the clustering.

**Table 8.** Item total statistics.

| | Variables | Scale Variance If Item Deleted | Corrected Item-Total Correlation | Cronbach's Alpha If Item Deleted |
|---|---|---|---|---|
| 1 | Enjoyability vs. other Labs | 292.45 | 0.577 | 0.886 |
| 2 | Familiarisation with MS Teams vs. other modules | 292.59 | 0.484 | 0.887 |
| 3 | Satisfaction with CAD I vs. other modules | 275.48 | 0.646 | 0.883 |
| 4 | Insecure about following | 293.20 | 0.392 | 0.889 |
| 5 | Comfortable for passing the finals | 298.15 | 0.267 | 0.891 |
| 6 | Affected by technical issues | 296.62 | 0.320 | 0.890 |
| 7 | Data consumption from downloading YouTube support channel videos | 288.99 | 0.413 | 0.888 |
| 8 | Assignment load | 299.96 | 0.279 | 0.890 |
| 9 | Hours of study during the week | 295.33 | 0.343 | 0.889 |
| 10 | Are you satisfied with your assignment grades | 304.09 | 0.177 | 0.891 |
| 11 | Module's organisation | 298.14 | 0.349 | 0.889 |
| 12 | How well tasks are assessed during online lectures | 280.07 | 0.554 | 0.885 |
| 13 | Did the theory contribute to the assignments? | 300.46 | 0.452 | 0.889 |
| 14 | Assignments' variety (quizzes, sketches, CAD drawings) | 293.02 | 0.487 | 0.887 |
| 15 | Class notes, download and evaluate | 288.15 | 0.367 | 0.890 |
| 16 | Did quizzes help the understanding of the theoretical part? | 295.50 | 0.484 | 0.888 |
| 17 | Have you conceived of the meaning of planes | 293.97 | 0.573 | 0.887 |
| 18 | Cutting planes highlighted in 3D views | 297.00 | 0.383 | 0.889 |
| 19 | Have you fully perceived the object | 300.67 | 0.319 | 0.890 |
| 20 | Quality of videos concerning the solving methodology | 302.54 | 0.234 | 0.891 |
| 21 | Clarity of video, image and sound | 284.40 | 0.508 | 0.886 |
| 22 | Social Media application skills | 300.44 | 0.346 | 0.889 |
| 23 | Classroom fatigue | 296.36 | 0.275 | 0.891 |
| 24 | Express out loud your questions | 281.03 | 0.572 | 0.885 |
| 25 | Questions being solved during synchronous lectures | 295.49 | 0.494 | 0.888 |
| 26 | Resentment when instructor does not return graded tasks on time | 291.37 | 0.574 | 0.886 |
| 27 | Comments by the instructor help the understanding of mistakes | 294.57 | 0.365 | 0.889 |
| 28 | Instructor helped meeting new people during synchronous lectures | 293.87 | 0.337 | 0.890 |
| 29 | Knowledge Weaknesses | 297.81 | 0.291 | 0.890 |
| 30 | Computer skills | 298.81 | 0.308 | 0.890 |
| 31 | Tasks assigned relevant to future work | 305.38 | 0.126 | 0.892 |
| 32 | Presentation and clarity of the 15th assignment | 282.90 | 0.387 | 0.891 |
| 33 | 15th assignment related to real-world tasks | 292.65 | 0.466 | 0.887 |
| 34 | All assignments related to real-world tasks | 298.70 | 0.293 | 0.890 |
| 35 | Overall evaluation of the module | 296.50 | 0.411 | 0.888 |
| 36 | Dealing with weaknesses and lack of knowledge | 293.77 | 0.496 | 0.887 |
| 37 | Are you likely to succeed in a similar future task | 271.75 | 0.738 | 0.881 |

**Table 9.** Cluster analysis table.

| | Variables | Cluster | | Anova |
|---|---|---|---|---|
| | | 1 | 2 | Sig. |
| 1 | Enjoyability vs. other Labs | 5 | 4 | 0.00 |
| 2 | Familiarisation with MS Teams vs. other modules | 4 | 4 | 0.00 |
| 3 | Satisfaction with CAD I vs. other modules | 9 | 7 | 0.00 |
| 4 | Insecure about following | 4 | 3 | 0.00 |
| 5 | Are you comfortable passing the finals | 4 | 3 | 0.02 |
| 6 | Affected by technical issues | 4 | 3 | 0.00 |
| 7 | Data consumption from downloading YouTube support channel videos | 5 | 3 | 0.00 |
| 8 | Assignment load | 4 | 3 | 0.00 |
| 9 | Hours of study during the week | 6 | 5 | 0.00 |
| 10 | Are you satisfied with your assignment grades | 4 | 4 | 0.28 |
| 11 | Module's organisation | 4 | 3 | 0.00 |
| 12 | How well tasks are assessed during online lectures | 9 | 7 | 0.00 |
| 13 | Did the theory contribute to the assignments? | 4 | 4 | 0.00 |
| 14 | Assignments' variety (quizzes, sketches, CAD drawings) | 4 | 3 | 0.00 |
| 15 | Class notes, download and evaluate | 3 | 2 | 0.00 |
| 16 | Did quizzes help the understanding of the theoretical part? | 4 | 4 | 0.00 |
| 17 | Have you conceived of the meaning of planes? | 4 | 4 | 0.00 |
| 18 | Cutting planes highlighted in 3D views | 4 | 4 | 0.00 |
| 19 | Have you fully perceived the object | 5 | 4 | 0.02 |
| 20 | Quality of videos concerning the solving methodology | 4 | 4 | 0.01 |
| 21 | Clarity of video, image and sound | 9 | 8 | 0.00 |
| 22 | Social Media application skills | 4 | 4 | 0.00 |
| 23 | Classroom fatigue | 5 | 4 | 0.01 |
| 24 | Express out loud your questions | 4 | 2 | 0.00 |
| 25 | Questions being solved during synchronous lectures | 5 | 4 | 0.00 |
| 26 | Resentment when instructor does not return graded tasks on time | 5 | 4 | 0.00 |
| 27 | Comments by the instructor help the understanding of mistakes | 4 | 3 | 0.00 |
| 28 | Instructor helped meet new people during synchronous lectures | 4 | 3 | 0.00 |
| 29 | Knowledge Weaknesses | 3 | 2 | 0.00 |
| 30 | Computer skills | 4 | 4 | 0.00 |
| 31 | Tasks assigned relevant to future work | 5 | 4 | 0.23 |
| 32 | Presentation and clarity of the 15th assignment | 8 | 7 | 0.00 |
| 33 | 15th assignment related to real world-tasks | 4 | 3 | 0.00 |
| 34 | All assignments related to real world-tasks | 4 | 4 | 0.00 |
| 35 | Overall evaluation of the module | 4 | 3 | 0.00 |
| 36 | Dealing with weaknesses and lack of knowledge | 4 | 3 | 0.00 |
| 37 | Are you likely to succeed in a similar future task? | 9 | 7 | 0.00 |

*3.5. Hierarchical Cluster Analysis*

Hierarchical cluster analysis is a method used to segregate the learners into clusters with successively increasing statistical proximity. The criterion for the segregating process is the similarity of their answers on the questionnaire. The techniques of variance analysis were implemented, and distances between the clusters were computed. This particular method was performed in several stages where the clusters were joined, and the sum of squares represents a loss criterion [19]. As a result of the procedure, a dendrogram was created, as shown in Figure 8, where in the upper branch, two large clusters are distinguished. Three clusters appear in the next branch, and five other clusters in the branch underneath.

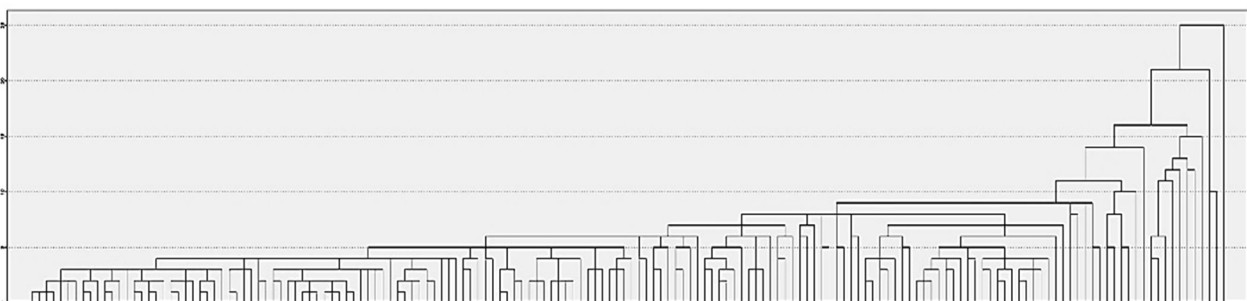

**Figure 8.** Tree diagram (differences between the group clusters).

## 4. Conclusions

Teaching materials, global connectivity, enhanced learner involvement, and communication have been fundamental elements of the e-learning philosophy [28]. During the emerging circumstances of the distance learning environments caused by the COVID-19 pandemic, most higher-education modules have used LMSs and communication platforms in order to fulfil the requirements of remote classrooms. In the case of the University of West Attica, in Athens Greece, MS Teams was used as a synchronous method of e-learning lecture transmission. Since most modules had already developed their course flow during the pre-pandemic period, using LMSs such as Moodle and E-class, both LMSs were applied simultaneously, and this "double" platform engagement was found to provide a fast-track response to the emerging needs of this new period of restrictions. Following an e-module via a parallel LMS and communication platform presented a high confusion risk to both instructors and learners, considering the absence of their physical presence.

In this paper, the use of a single learning platform [29] was tested, using as a testbed an online mechanical CAD module, addressed to first-semester university-level students. Students are facing the impact of the long-term limitations [30] related to COVID-19 ERTE, and especially first-year university students [31] for whom the adjustment inside the faculty environment is crucial during the first semester; otherwise, high dropout rates can be observed during this period [20].

The initial challenge of this study was to avoid any disruption in knowledge transmission to the youngest age group of the academic community. Integrating tasks' assignment into a single platform that may be utilised for synchronous lectures at the same time a providing asynchronous support needed to be tested and evaluated.

To achieve the above, the importance of several parameters affecting students' attitudes towards the task assignments needed to be tested. The use of electronic means facilitated the collection of data, as well as their indexing and management. Online surveys, additional student information, and platform reports (insight add-on) were gathered and merged into a matrix, and after being statistically analysed, 37 variables corresponding to the survey questions were found to be significantly related to first-year students' attitudes towards the assignments required by the e-module attendance during their first semester in the School of Mechanical Engineering.

Cronbach's alpha coefficient was computed, and it came out to be 0.891, which is considered good, demonstrating that 89% of the questionnaire was considered internally consistent and reliable for future experiments in a similar field. The cluster analysis distinguished two main learner groups, based on learners' commitment to the e-module, their competencies, and their level of active learning, allowing them to feel competent and confident to complete a similar project in the future.

Amongst the most interesting findings of this research is that the use of a single online platform for real-time lecture transmission and task assessments, class notes, and grading, as well as links for asynchronous support, by integrating coursework and learning materials' links posted on social media channels, made task organisation more convenient for both learners and lecturers.

The renewed agenda for the development of higher education policies in EU countries has identified four key goals; among them, there is the innovation contribution of higher education establishments, as well as the support of effective learning systems. Blended learning environments with similar learning frameworks could be incorporated into the learning procedure, since they have been tested and found adequate to benefit the academic community.

## 5. Future Work

Considering the current pandemic circumstances, the effects of the imperative return to face-to-face learning environments in universities where physical distancing measures have been overlooked are yet to be evaluated. VLEs could respond to similar global health situations, especially when the remote learning and asynchronous support part of the knowledge transmission have been already tested, in strictly distance learning environments. Blended learning environments could also benefit from the above outcomes.

As universities around the world have already proposed hybrid solutions for the academic year 2021–2022 [32], future work could include the application of the specific methodology in blended learning environments [33], by substituting the "assignment" feature of a typical face-to-face classroom with the one, practiced during the pandemic. The communication platform could work in current teaching spaces as a parallel online module for transmission including all assignments and support features performed in the present experiment.

The 37 variables discovered in this study can be further evaluated under virtual teaching spaces, and the outcomes extracted could benefit the educational community, especially during this intermediate period where the pandemic risks have not yet been overcome.

**Author Contributions:** Conceptualisation, Z.K. and C.S. (Constantinos Stergiou); methodology, Z.K.; software, G.B.; validation, Z.K. and C.T.; formal analysis, G.B.; investigation, Z.K.; resources, Z.K.; data curation, Z.K.; writing—original draft preparation, Z.K.; writing—review and editing, C.S. (Constantinos Stergiou), C.T.; visualisation, Z.K.; supervision, C.S. (Constantinos Stergiou), C.S. (Cleo Sgouropoulou); project administration, C.S. (Constantinos Stergiou). All authors have read and agreed to the published version of the manuscript.

**Funding:** This research received no external funding.

**Data Availability Statement:** The data used to support the findings of this study have not been made available because they contain information that could compromise research participant privacy/consent.

**Acknowledgments:** The authors would like to acknowledge the University of West Attica, the Department of Mechanical Engineering and the Department of Informatics and Computer Engineering, for supporting this research.

**Conflicts of Interest:** The authors declare no conflict of interest.

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
