# Peer review of "Evaluating Remote Task Assignment of an Online Engineering Module through Data Mining in a Virtual Communication Platform Environment"

_electronics, doi:10.3390/electronics11010158_

Round 1

Reviewer 1 Report

Dear authors,

congratulations for your work. 

After reading your article, I make the following recommendations:

  • Although figure 5 describes the method of data collection, I consider that it is relevant to include a subsection where the sample data is described and the descriptive statistics are presented. It is extremely important to understand the characteristics of the sample, such as the percentage of girls in the sample, the average entrance grade, among other information.

  • The results are demonstrated, but the results discussion must be improved. The conclusions are interesting; however, they could be further explored, particularly with regard to conclusions at the level of educational policies.

Best regards,

Author Response

Reviewer 1

1

Although figure 5 describes the method of data collection, I consider that it is relevant to include a subsection where the sample data is described and the descriptive statistics are presented. It is extremely important to understand the characteristics of the sample, such as the percentage of girls in the sample, the average entrance grade, among other information.

A sub chapter “Data sample characteristics: has been added:

In order to understand the characteristics of the sample under the specific context of the pandemic, demographic factors and additional social aspects have been included in the questionnaire, as described below:

Out of the 165 participants, most of them were aged between 18-21 years (87%). 11% were between 22-25 years old and 2% 26-29 years old and 1% 34-37 years old. The majority were male students (88%) and the remaining 12% female students [16]. 64% were attending the e-lectures from the same town, 34% from other cities or provinces and 2% from other countries. 77% were living in the same dwellings with their families, 14% were living by themselves, 4% with their siblings, 4% with their companions and the remaining 1% were living together with a friend. In June 2020, students have passed the University entrance exams, with average grade in mathematics 13.90, in dissertation 13.20, in chemistry 12.40 and in physics 13.95. As regards to the selection of the University Department of Mechanical Engineering, 64% stated that it was their first choice, versus 33% which have selected the department due to the proximity of the campus with their place of residence. The remaining 3% stated that their department selection has been based to other criteria

2

The results are demonstrated, but the results discussion must be improved. The conclusions are interesting; however, they could be further explored, particularly with regard to conclusions at the level of educational policies.

Added in the section :conclusions”: The renewed agenda for the development of higher education policies in EU countries has identified four key goals, among them the innovation contribution of higher education establishments, as well as the support of effective learning systems. Blended learning environments with similar learning frameworks could be incorporated in the learning procedure, since they have been tested and proven to benefit the academic community.

Reviewer 2 Report

The problems associated with this paper are:

  1. The (a) Number of participant were : 216, (b) Subpopulation : 190, and (c) Valid scores : 165. How do the authors come about this? Kindly explain.
  2. Please improve the English writing of the paper.

Author Response

Reviewer 2

1

English language and style are fine/minor spell check required

The manuscript has been checked and all the mistakes were corrected

2

The (a) Number of participant were: 216, (b) Subpopulation: 190, and (c) Valid scores : 165. How do the authors come about this? Kindly explain.

216 students have initially registered in the online module, divided in 11 online teaching groups. 190 students out of 216 participated in the survey. Out of those 190 participants, 165 scores have been retained and considered valid for this research, since the remaining 25 scores reflected to older attendants, not first year students.  

A sub chapter “Data sample characteristics” has also been added

3

Please improve the English writing of the paper.

The manuscript has been checked by a native English-speaking colleague in order to revise and improve the English language.

Reviewer 3 Report

In order to cope with the epidemic outbreak and meet the needs of the current era, the author puts forward 37 variables and designs an online learning platform. The author has a good starting point, but there are still some shortcomings and shortcomings, as shown below;

  1. The summary does not specify the specific problems to be solved. It is suggested to reorganize the summary, for example, what are the potential obstacles.
  2. The introduction of the first part is too casual and the planning is not reasonable. It is suggested to summarize it again.
  3. The method of the second part is too general, and more specific contents are suggested.
  4. Please introduce the selection basis of 37 variables in the author's paper.
  5. Please introduce the selection criteria of threshold in the author's paper.
  6. It is suggested that the author can compare with more platforms.
  7. The following article may be instructive to the author, please read it carefully.

Multipopulation optimization for multitask optimization

Multi-objective optimization for location-based and preferences-aware recommendation

Author Response

Reviewer 3

1

Moderate English changes required

The manuscript has been checked and corrected by a native English-speaking colleague.

2

The summary does not specify the specific problems to be solved. It is suggested to reorganize the summary, for example, what are the potential obstacles.

Several parts of the abstract have been corrected. Also, it was enriched in terms of the novelty of the research and findings. As for the potential obstacles, we included a new section (“Difficulties encountered during the research”) in which they were further analysed.  We would like to note that there is a limit in the words of abstract, which has been exceeded since we followed the reviewer’s comment.

3

The introduction of the first part is too casual and the planning is not reasonable. It is suggested to summarize it again.

The introduction was improved. We corrected several parts, we added new references and also we added on new paragraph.

4

The method of the second part is too general, and more specific contents are suggested

 A subchapter “Difficulties encountered during the research” has been added in the data filtering methods

Several difficulties have been encountered during this research. Both questionnaires were identified and some students that have filled the pre-course survey quit the module before having the opportunity to participate on the post course questionnaire. In addition to that matter, the second survey contained multiple variables aiming to investigate several constructs. The total number of questions of the second survey was 109 and the average time to complete was 34:56, which implied a significant level of difficulty. Both surveys needed to be fused in one, by incorporating data from students’ registry, one by one. Nevertheless, students have been overall highly motivated to participate on both surveys, probably due to the lockdown measures and the students’ need to express their opinion and attitude towards the specific learning method.

5

Please introduce the selection basis of 37 variables in the author's paper.

In the section “Data mining sources”: Among those were students’ registry data, that provided information about students’ previous educational performance, such as their grades in the university level entrance exams in the four subjects: physics, mathematics, chemistry and dissertation. Each student high school type has been considered, between general lyceum and vocational lyceum. Students’ activity reports were provided by the MS Teams communication platform, which included a median of their assignments’ grade, virtual presence, and asynchronous presence in the platform [15-17].

The purpose of this questionnaire was to investigate the impact of several constructs related to the online learning process during the pandemic, e.g. demographic factors, technical difficulties, the impact of social distancing, e-course presentation and computer skills, on students’ learning achievements [13]. Among other constructs, the construct of “assignments” was identified through 37 survey questions [16]. All answers have been associated with the registry number of each student, in order to link their attitude towards the online module with their past academic performance

6

Please introduce the selection criteria of threshold in the author's paper

For each statistical analysis performed, the criteria of threshold has been mentioned, for example:

A threshold of >0.20 was set in the calculation of Spearman’s rho coefficient

Cronbach’s alpha usually ranges from zero to one,

0.891 is considered a good measurement of reliability

Kaiser-Meyer-Olkin test, and the Bartlett's test of sphericity

The result from Bartlett's test entails that the highest significance (0.000) is attained

Anova field shows the level of the variables’ significance, which in most of them is <0.05 was given a value equal to 0.816, proving that the sample has a good adequacy.

7

It is suggested that the author can compare with more platforms.

The experiment took place during the first semester of 2020-2021, based on the learning platform applied by the guidelines of the University of West Attica for all schools and faculty members.   A comparison with other platforms cannot be done at this moment since at the time the research took place, there was no other option.

8

The following article may be instructive to the author, please read it carefully.

Multipopulation optimization for multitask optimization

Multi-objective optimization for location-based and preferences-aware recommendation

We have read both articles which were very interesting and we decided to include them in the Introduction section of our manuscript.

Round 2

Reviewer 3 Report

The author should adder the recently proposed papers, and revise the grammar carefully.

Author Response

We would like to thank the Reviewer for the comment.
However, the proposed articles have already been included in the revised manuscript from the first review round.
Please check the reference list in the manuscript and see that we incorporated your proposed articles:

  • Ref. 11. Wang, S.; Gong, M.; Wu, Y.; Zhang, M. “Multi-objective optimization for location-based and preferences-aware recommendation”, Information Sciences, Volume 513, 2020, pp. 614-626, ISSN 0020-0255, doi.org/10.1016/j.ins.2019.11.028.
  • Ref. 12. Tang, Z.; Gong, M.; Jiang, F.; Li, H.; and Wu, Y. "Multipopulation Optimization for Multitask Optimization," 2019 IEEE Congress on Evolutionary Computation (CEC), 2019, pp. 1906-1913, doi: 10.1109/CEC.2019.8790234.

Also, we are really impressed that you changed your point of view concerning the English language and style. Specifically, in the first review round, your decision was “Moderate English changes required” while in this second round, your decision is “Extensive editing of English language and style required”. We need to underline that in the first review round, the manuscript was checked by an English native speaker and all the mistakes, that were found, were corrected (you can see the corrections in the manuscript with the track & changes option). In this second round, we checked again the whole manuscript for mistakes. You can see how we corrected slight issues in the manuscript with the track & changes option.
Please note that concerning the English language and style, the decision of Reviewer 2 is “English language and style are fine/minor spell check required”.
Thank you for your time to review our work.

We hope that this new revised version of our paper meets your expectations.
